# A naturalistic study of brushing patterns using powered toothbrushes

**Mahmoud Essalat**[1]*, **Douglas Morrison**[2], **Sumukh Kak**[3], **E. Jun Chang**[4], **Isabel Roig Penso**[3], **Rachel J. Kulchar**[5], **Oscar Hernan Madrid Padilla**[6], **Vivek Shetty**[7]

1 Department of Electrical and Computer Engineering, University of California, Los Angeles, California, United States of America, 2 Department of Biostatistics, University of California, Los Angeles, California, United States of America, 3 Undergraduate, University of California, Los Angeles, California, United States of America, 4 Herman Ostrow School of Dentistry, USC, Los Angeles, California, United States of America, 5 Undergraduate, Princeton University, Princeton, New Jersey, United States of America, 6 Department of Statistics, University of California, Los Angeles, California, United States of America, 7 School of Dentistry, University of California, Los Angeles, California, United States of America

* mahmoudessalat@ucla.edu

**Data Availability Statement:** The dataset used as well as the code to generate the results are publicly available publicly at a GitHub repository. Essalat, M., & Morrison, D. (2022). Brushing-Behavior

## Abstract

Dental caries and periodontal disease are very common chronic diseases closely linked to inadequate removal of dental plaque. Powered toothbrushes are viewed as more effective at removing plaque; however, the conflicting evidence and considerable unexplained heterogeneity in their clinical outcomes does not corroborate the relative merits of powered tooth brushing. To explain the heterogeneity of brushing patterns with powered toothbrushes, we conducted a observational study of tooth brushing practices of 12 participants in their naturalistic setting. Integrated brush sensors and a digital data collection platform allowed unobtrusive and accurate capture of habitual brushing patterns. Annotated brushing data from 10 sessions per participant was chosen for scrutiny of brushing patterns. Analysis of brushing patterns from the total 120 sessions revealed substantial between- and within-participant variability in brushing patterns and efficiency. Most participants (91.67%) brushed for less than the generally prescribed two minutes; individual participants were also inconsistent in brushing duration across sessions. The time devoted to brushing different dental regions was also quite unequal. Participants generally brushed their buccal tooth surfaces more than twice as long as the occlusal (2.18 times longer (95% CI 1.42, 3.35; p < 0.001)) and lingual surfaces (2.22 times longer (95% CI 1.62, 3.10; p < 0.001); the lingual surfaces of the maxillary molars were often neglected (p < 0.001). Participants also varied in the epochs of excessive brushing pressure and the regions to which they were applied. In general, the occlusal surfaces were more likely to be brushed with excessive pressure (95% CI 0.10, 0.98; p = 0.015). Our study reveals that users of powered toothbrushes vary substantially in their use of the toothbrushes and diverge from recommended brushing practices. The inconsistent brushing patterns, between and within individuals, can affect effective plaque removal. Our findings underscore the limited uptake of generic oral self-care recommendations and emphasize the need for personalized brushing recommendations that derive from the objective sensor data provided by powered toothbrushes.

(Version 1.0.0) [Computer software]. https://doi.
org/10.5281/zenodo.1234.

**Funding:** Research reported in this publication was
supported by the National Institute of Dental and
Craniofacial Research under award number
R01DE025244 and 1UG3DE028723-01. The
content is solely the responsibility of the authors
and does not necessarily represent the official
views of the National Institutes of Health.

**Competing interests:** The authors declare no
conflict of interest.

## Introduction

Dental caries and periodontal disease are very common chronic diseases closely linked to inad-
equate oral self-care [1,2]. Scientific evidence indicates that regular and systematic toothbrush-
ing prevents the accumulation of dental plaque that leads to gum disease, tooth decay, and
eventually, tooth loss [3,4]. Thus, most efforts to reduce the incidence and impact of dental dis-
ease focus on toothbrushing techniques and strategies that reduce the accumulation of plaque.
Although manual toothbrushes are the most commonly used tools for plaque control, powered
toothbrushes are gaining increasing acceptance as alternatives, especially for children, people
with disabilities or limited mobility, and older adults. Automated powered toothbrushes mini-
mize the manual efforts of brushing and incorporate timers to reinforce brushing duration;
thus, they are viewed as more effective at removing plaque than manual brushes [5]. Despite a
substantial heterogeneity ($I(2) > 80\%$) in plaque removal, the Cochrane meta-analyses found
that powered toothbrushes produced an 11% reduction in plaque at one to three months of
use and a 21% reduction in plaque after three months of use [5].

The considerable heterogeneity in plaque removal with powered toothbrushes ($I(2) > 80\%$)
observed in the Cochrane study [5], could not be explained by the difference in types of pow-
ered toothbrushes and more likely derives from by varying brushing patterns. For toothbrush-
ing to be effective, all dental surfaces need to be cleaned frequently and adequately; otherwise,
the practical value of toothbrushing is low. Brushing frequency, duration, and technique are
key determinants of adequate plaque reduction and optimal oral self-care.

Brushing techniques commonly recommended by dental professionals [6] are based on
manual toothbrushes [6,7]. They do not readily translate to brushing with powered tooth-
brushes where the user guides but does not animate the brush head. Studies that have
attempted to clarify brushing patterns with powered toothbrushes have largely focused on
between-individual variations determined through video recordings obtained in controlled
clinical settings [8]. Such snapshot observations ignore within-participant variations in brush-
ing behaviors and patterns across multiple sessions. Moreover, the external validity of the
assessments in simulated environments suffers because brushing behavior patterns recorded
in research settings may not reflect naturalistic, real-world practices.

To explain the heterogeneity of brushing patterns with rotation-oscillation powered tooth-
brushes, we conducted an observational study of individuals in their home settings. Sensors
embedded within the powered toothbrush and concurrent video recordings of brushing ses-
sions ensured data fidelity and ecological validity across multiple days. Our objective was to
gather accurate data on habitual brushing patterns using powered toothbrushes, with a focus
on the duration of each session, tooth surfaces covered, and episodes of excessive brushing
pressure per session. By examining habitual brushing patterns at the individual and session-
level, we sought to clarify between-person and session-to-session variability in brushing pat-
terns and efficiency using powered toothbrushes.

## Materials and methods

This study comprised part of a larger study involving machine learning approaches to charac-
terize toothbrushing behaviors and develop a brushing efficiency score. As part of the parent
study, 12 healthy college students with no evident dental disease, provided their brushing data
in the home setting over three weeks (50 sessions each). Basic instruction on the use of the
brush and setting up the data collection system was given and the participants were instructed
to freely brush their teeth in a manner most natural to them. All participants provided written
informed consent and the study protocol was reviewed and approved by the Institutional
Review Board of the University of California, Los Angeles (IRB#18–000874).

## Data collection infrastructure

To allow objective, individual-level, and ecologically-valid data on oral hygiene behaviors, we deployed the Remote Oral Behaviors Assessment System (ROBAS) described previously [9]. Briefly, ROBAS builds on a broadly available consumer-grade powered toothbrush (Oral-B Genius X; Procter & Gamble) as the data source for brushing behaviors (timing, duration, pressure applied). The Oral-B brush employs a rotational-oscillation mode of action and brushing movements are captured by an accelerometer, gyroscope, and pressure sensor contained within the powered brush. Captured data is transmitted over BLE (Bluetooth Low Energy) to a paired smartphone running the companion data collection app. Collected data is then uploaded to a secure cloud server for remote monitoring of data yields and analytics. Visualization of time-series data streams of brushing episodes and remote monitoring of sensor function and participant compliance is accomplished through an adaptation of the open platform Grafana™ dashboard [10].

## Data collection

Upon enrollment, each participant was provided an Oral-B powered toothbrush, a suction-cup phone mount, charger, and quick-start instructions. Participants downloaded the study-specific app onto their own smartphone and paired it to the powered brush. Participants were instructed on the operation of the powered brush and on how to mount the smartphone to their bathroom mirror during a brushing session for the duration of the study (3 weeks). Participants were instructed to brush twice daily for two minutes each time. At the start of each brushing session, participants launched the study app and activated the smartphone camera. Data from the embedded sensors was buffered by the brush and transmitted to the study phone via Bluetooth. Turning off the toothbrush ended data collection and triggered the app to save the timestamped brushing data. The brushing session data and the corresponding videoclip were then uploaded via the ROBAS platform to a secure cloud server for subsequent analysis. The ROBAS platform with integrated Grafana™ dashboard allowed research staff to remotely monitor data feeds and conduct quality checks.

## Data processing and annotation

We selected 10 sessions (out of the 50 recorded sessions per participant) at random for a total of 120 brushing sessions across the 12 participants. To establish ground truth information, trained and calibrated researchers reviewed and annotated the individual video recordings, focusing on tooth surface coverage as well as brushing duration of each surface. Every epoch of brushing a dental region that lasted more than 0.5 second was labeled by marking the beginning and end timestamp of that brushing epoch. Because the study phone camera recorded at 1080 p at 30 fps (frames-per-second), we were able to get a 33 ms time resolution for the ground truth data. The annotated video timestamps were then aligned with the sensor signal time stamps. An experienced examiner conducted random audits of labeled data and provided corrections on annotation errors.

Dental regions (Fig 1) were characterized using the convention proposed by Lee et al. [11]. Briefly, dental regions are labeled by jaw (maxilla or mandible), side (right, anterior or left), and tooth surface (buccal, lingual, or occlusal).

## Analyses

All statistical analysis was performed in R version 4.1.1 [12], using the regression modeling package "glmmTMB" [13]. The boxplots are generated using Matlab R2021a [14]. The dataset

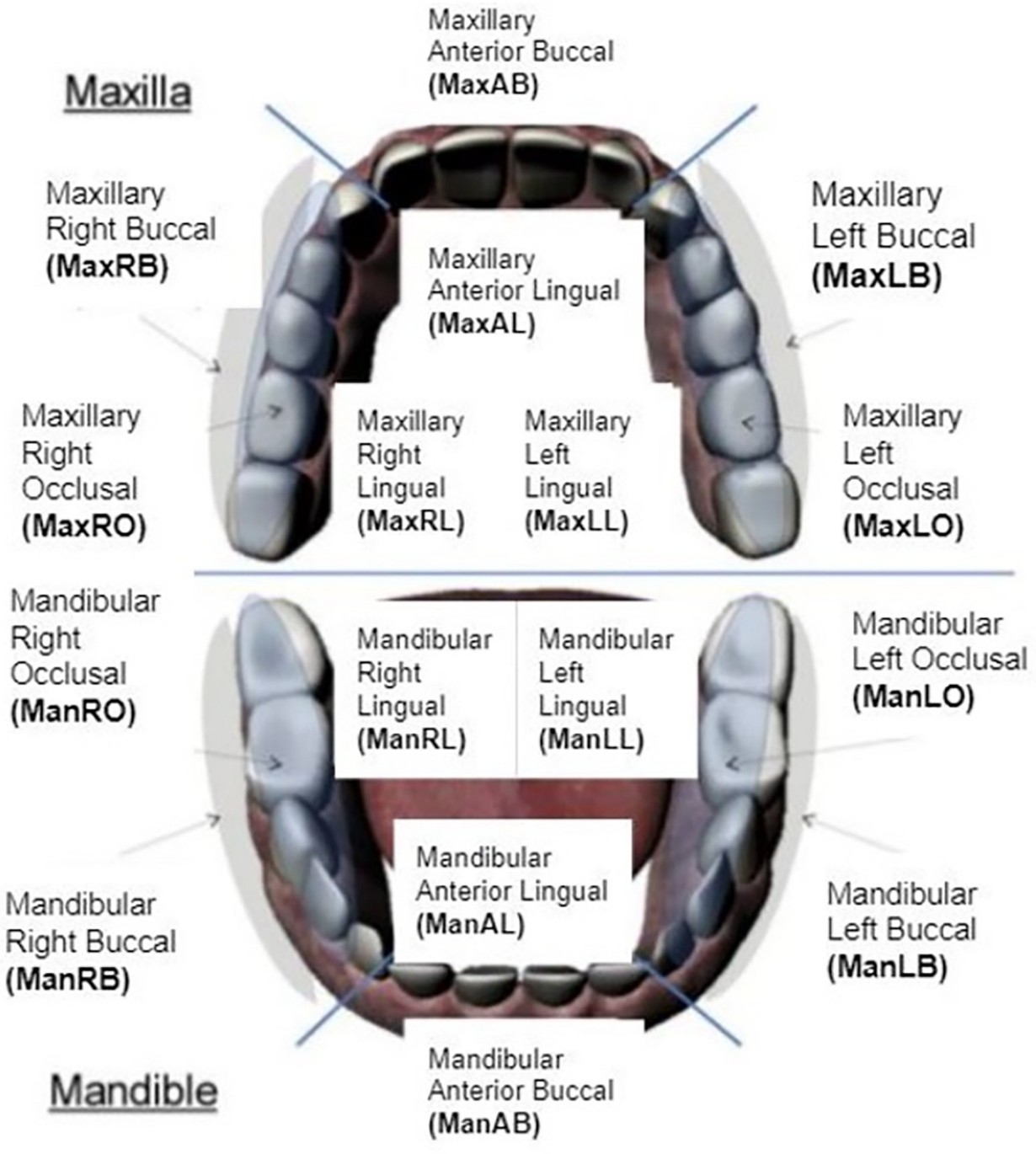

**Fig 1. 16 dental surfaces considered in this study (image retrieved from [11]).**

used as well as the code to generate the results are publicly available publicly at a GitHub repository [15].

Several participants skipped some regions altogether in some or all brushing sessions, and most participants avoided using excessive pressure most of the time. Therefore, the durations of region-specific brushing and region-specific brushing with excessive pressure were often

equal to zero, a statistical phenomenon referred to as zero-inflation [13]. Accordingly, we analyzed these outcomes using statistical models that account for zero-inflation. Specifically, we modeled these outcomes (each measured in counts of 25Hz samples) using zero-inflated generalized linear mixed-effects regression models, with a log-link and a negative binomial outcome distribution for the count submodels and a logistic link and a Bernoulli outcome distribution for the zero-inflation submodels. Both submodels had fixed effects for tooth surface, mouth side, and jaw, and to capture between- and within-participant variabilities, had random effects on the intercept by session nested in participant, to account for person-to-person and session-to-session differences in overall brushing duration, and person-specific overdispersion parameters to account for person-to-person differences in residual variance. The count submodel for region-specific brushing duration also included participant-specific random effects for tooth surface, mouth side, and jaw (see S1 Appendix A1 for details); the count submodel for region-specific excessive pressure duration failed to converge when random effects were added for these covariates, since many participants never brushed with excessive pressure in some or all regions.

We calculated the total active brushing duration per-session by excluding pauses in brushing and the epochs of transitioning the brush head to different dental surfaces. We modeled this outcome using a negative binomial generalized linear mixed-effects regression model (not zero-inflated, since all brushing sessions had duration greater than zero) with random intercepts by participant and fixed effects for participant-specific dispersion (see S1 Appendix A3 for details).

To examine the data on different levels, we used boxplots. Data points were labeled as outliers if they were not in the range of [q1—w * (q3-q1), q3 + w * (q3-q1)]; in which w is the Whisker value and q1 and q3 are the 25th and 75th percentiles of the sample data, respectively. We used a whisker value of $\pm 2.7\sigma$ ($\sigma$ is the standard deviation of the sample data) that corresponds to the coverage of 99.3% of the data, if the data is normally distributed. The significance-level () was set at 5%.

## Results

The 12 participants comprised of eight females and four males with ages ranging from 18 to 23 years (20.77 ± 1.59).

### Total active brushing duration in each session

We calculated the active brushing duration by excluding pauses in brushing and the times transitioning the brush head to different regions. Fig 2 summarizes the active brushing duration for all participants. Most of the participants (91.67%) brushed less than the prescribed two minutes in all sessions. The mean brushing duration for a participant was 89.22 seconds.

There was substantial between- and within- individual variability in brushing duration. Observed between-participant variability was 16.69 seconds (see Table A3.2.5 in S1 Appendix; coefficient of variation = 0.19 > 0.05) and observed within-participant variability was 14.50 seconds (see Table A3.2.5 in S1 Appendix; coefficient of variation = 0.17 > 0.05). Some participants (e.g. # 1 and 2) brushed for almost two minutes in most sessions, whereas others (e.g. participant 5) brushed for less than a minute. Some participants (e.g., 2, 4, and 7) brushed consistently for nearly the same duration each time; others (e.g., # 5, 9, and 12) varied greatly (> 70 seconds) in their brushing duration.

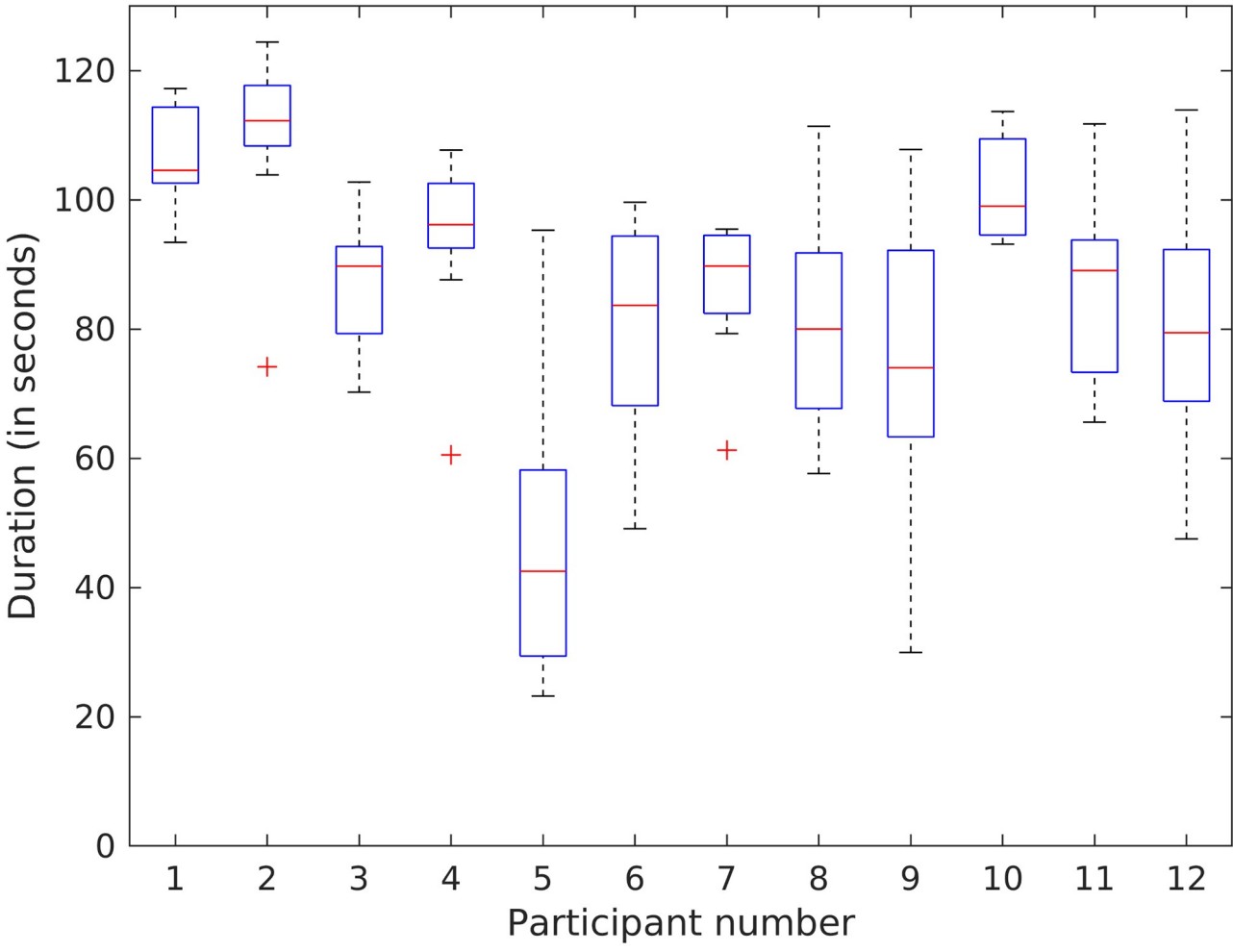

**Fig 2. Brushing durations for the 12 participants.** Active brushing duration is calculated by excluding pauses in brushing and the times transitioning the brush head to different regions.

### Brushing duration for each dental surface

Fig 3 summarizes the brushing duration of each region for all participants. MaxAB, MaxLB, and MaxRB were the areas brushed the longest with a median of 10.68, 8.78, and 8.22 seconds respectively. In contrast, MaxLL and MaxLO were frequently skipped during brushing.

Brushing time categorized by different regions is shown in Fig 4. Participants did not vary significantly in the brushing times spent on the maxillary and mandibular regions or different sides (right, anterior, and left). However, participants differed in the times spent brushing various teeth surfaces with buccal surfaces brushed significantly more than the lingual and occlusal surfaces. On average, buccal surfaces were brushed 2.18 times longer than the lingual surfaces (see Table A1.2.1 in S1 Appendix; 95% CI 1.42, 3.35; $p < 0.001$) and 2.22 times longer than the occlusal surfaces (see Table A1.2.1 in S1 Appendix; 95% CI 1.62, 3.10; $p < 0.001$).

### Between-individual variability in brushing duration

There was considerable between-individual variability in terms of brushing time devoted to different regions (Fig 5). Coefficient of variation for all the regions brushed during a session

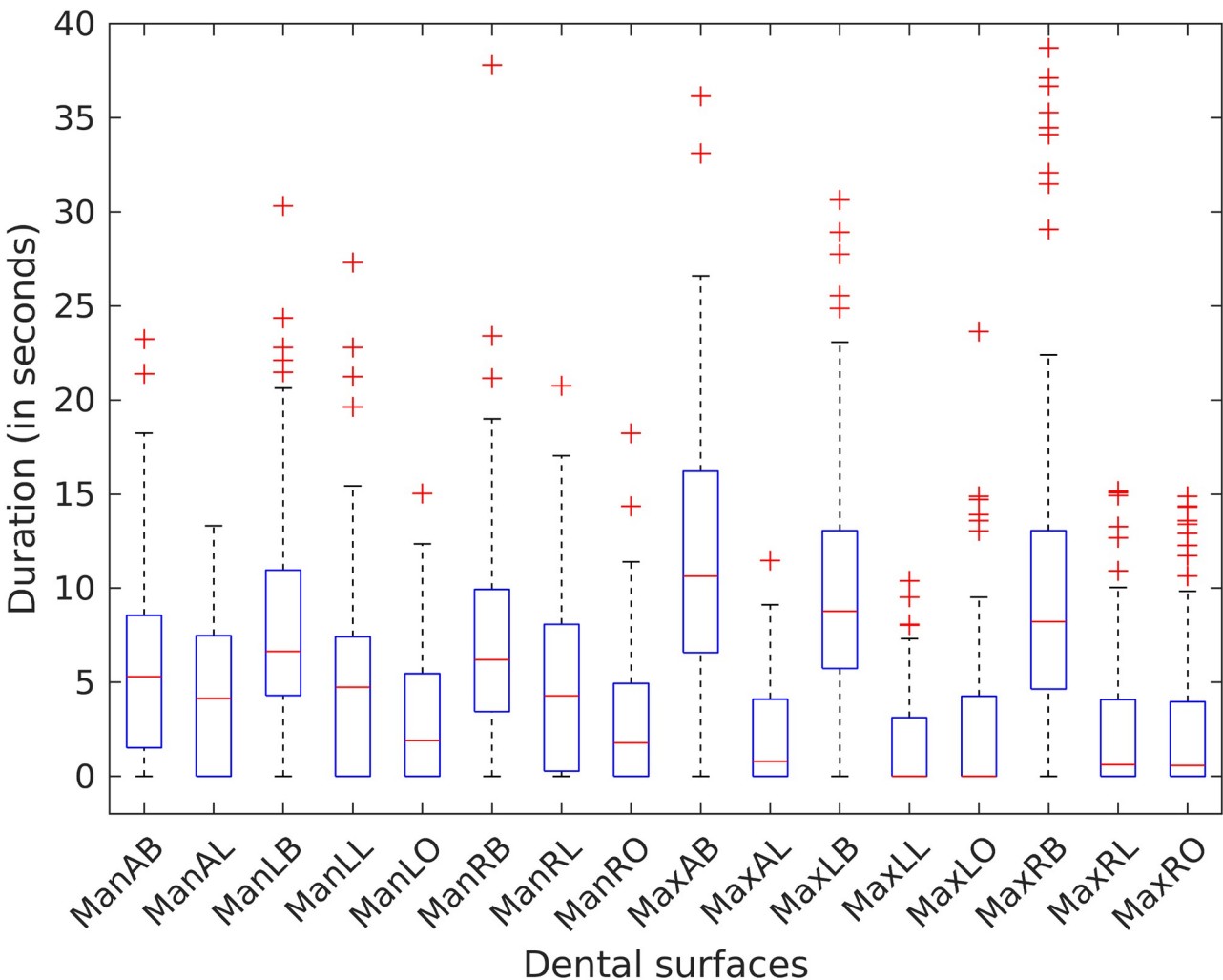

**Fig 3. Group-level brushing time of all dental surfaces.** MaxAB, MaxLB, and MaxRB were the areas brushed the longest and in contrast, MaxLL and MaxLO were frequently skipped during brushing.

was greater than 20%. (see S1 Appendix A2). Some (e.g. participant 7) brushed their maxillary teeth much more than their mandibular teeth (see Table A1.2.6 in S1 Appendix; p < 0.001). Others (e.g. participant 11), paid less attention to their lingual surfaces and focused primarily on the buccal surfaces (see Table A1.2.6 in S1 Appendix; p = 0.001).

## Within-individual variability in brushing duration

Participants varied greatly in their brushing of different regions across their brushing sessions. As exemplified by the brushing patterns of participant 2, the brushing duration for the lingual surfaces across sessions varied from none to 60 seconds (Fig 6).

## Episodes of excessive brushing pressure

Fig 7 summarizes the episodes of excess brushing pressure by region. About 16.7% of the participants exerted excessive brushing pressure of more than one second duration during each brushing session. Also, the occlusal surfaces were most frequently brushed with excessive

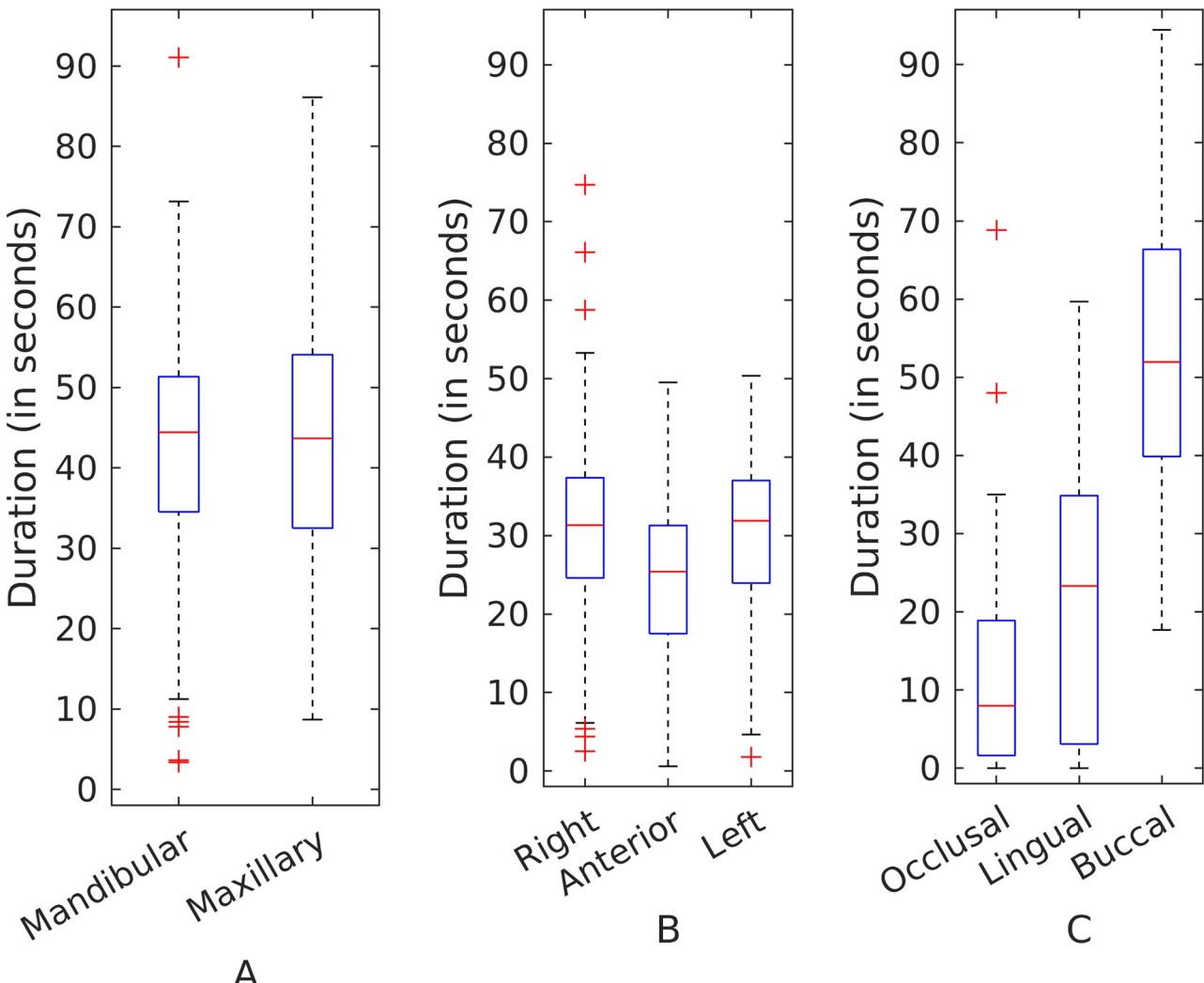

**Fig 4. Group-level brushing times of different dental regions.** Categorized by (A) jaw, (B) side, and (C) surface. Buccal surfaces were brushed twice more than the lingual and occlusal surfaces.

pressure (see Table A2.2.1 in S1 Appendix; estimated log relative duration = 0.54; 95% CI 0.10, 0.98; p = 0.015).

## Discussion

Our study revealed that brushing patterns with powered toothbrushes in the home setting varied greatly between individuals as well as within individuals. Although the electronic brushes incorporated timers and study participants were aware that their sessions were being monitored, most participants brushed their teeth for less than the prescribed two minutes. Even individual participants were inconsistent in the total duration of time they spent brushing over different days. The times devoted to brushing different dental regions are quite unequal at the individual level with certain regions receiving more attention than the others. Participants generally brushed the buccal tooth surfaces twice as long as the occlusal and lingual surfaces; the lingual surfaces of the maxillary molars were often neglected.

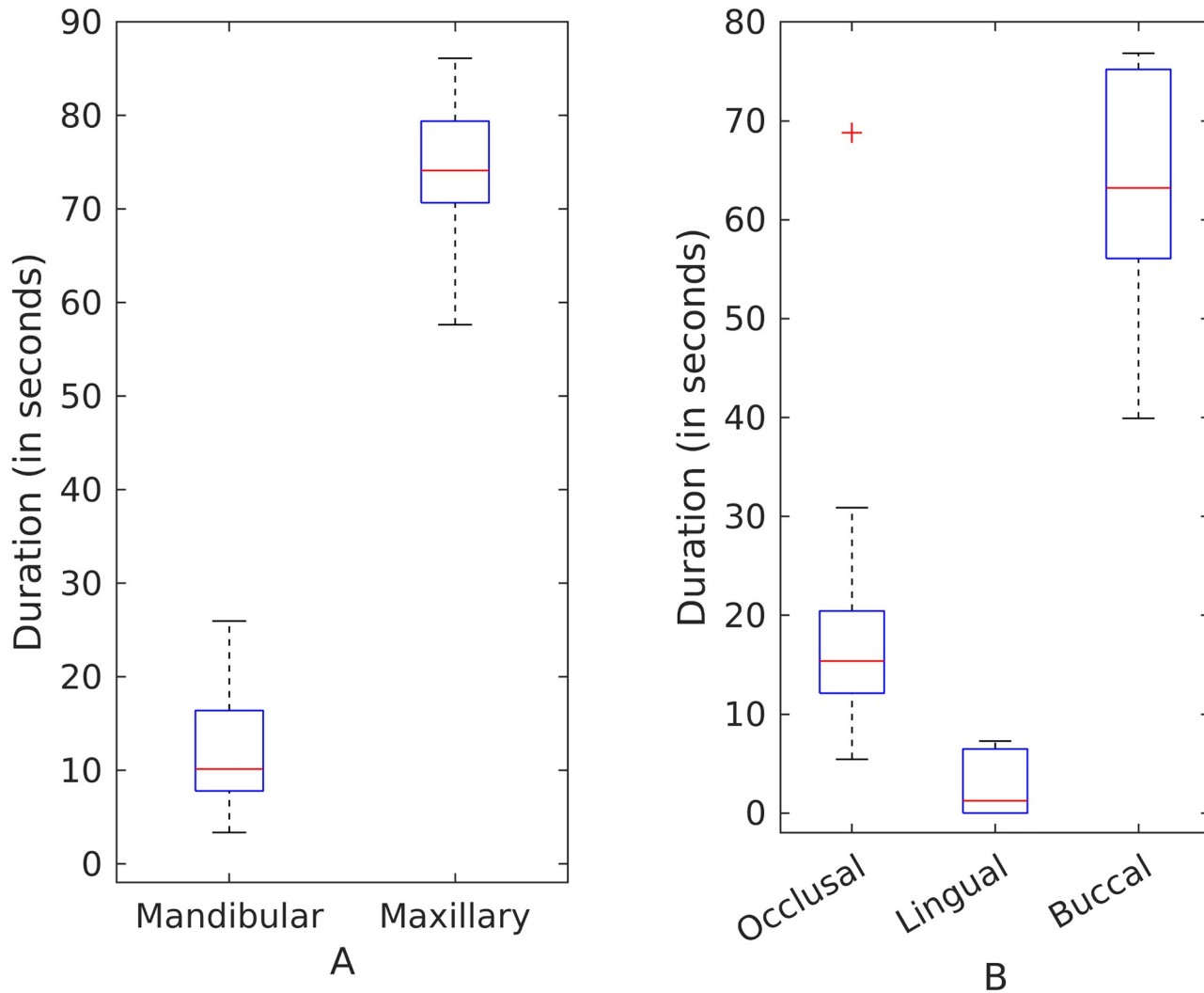

**Fig 5. Between-individual variability in brushing duration by region.** (A) by jaw and (B) by tooth surface.

The varying brushing patterns and unequal coverage of dental surfaces has implications for systematic and thorough brushing behavior. There are manifest discrepancies between professional recommendations on brushing techniques and what is actually practiced in the home setting. Our study participants were aware that their brushing behaviors were being monitored remotely; yet, they rarely brushed for the entire 120 seconds recommended by dentists. Most spent less than 85 seconds per brushing session, a finding consistent with other studies that indicate that patients often fail to adhere to the recommended brushing time of two minutes [16]. Increased brushing time is linked to better plaque removal and Gallagher et al [17] showed that brushing with manual brushes for 120 seconds removed 26% more plaque than brushing for 45 seconds. Despite the putative advantage, integrated timers alone may not facilitate adherence to recommended brushing times and need to be reinforced by other ways of engaging individuals.

Beyond the differences in total brushing times with the powered toothbrushes, the variability in the time devoted to brushing each region suggested inefficient brushing practices. Irrespective of the mode of action of the powered brush (i.e., rotating-oscillating or reciprocating),

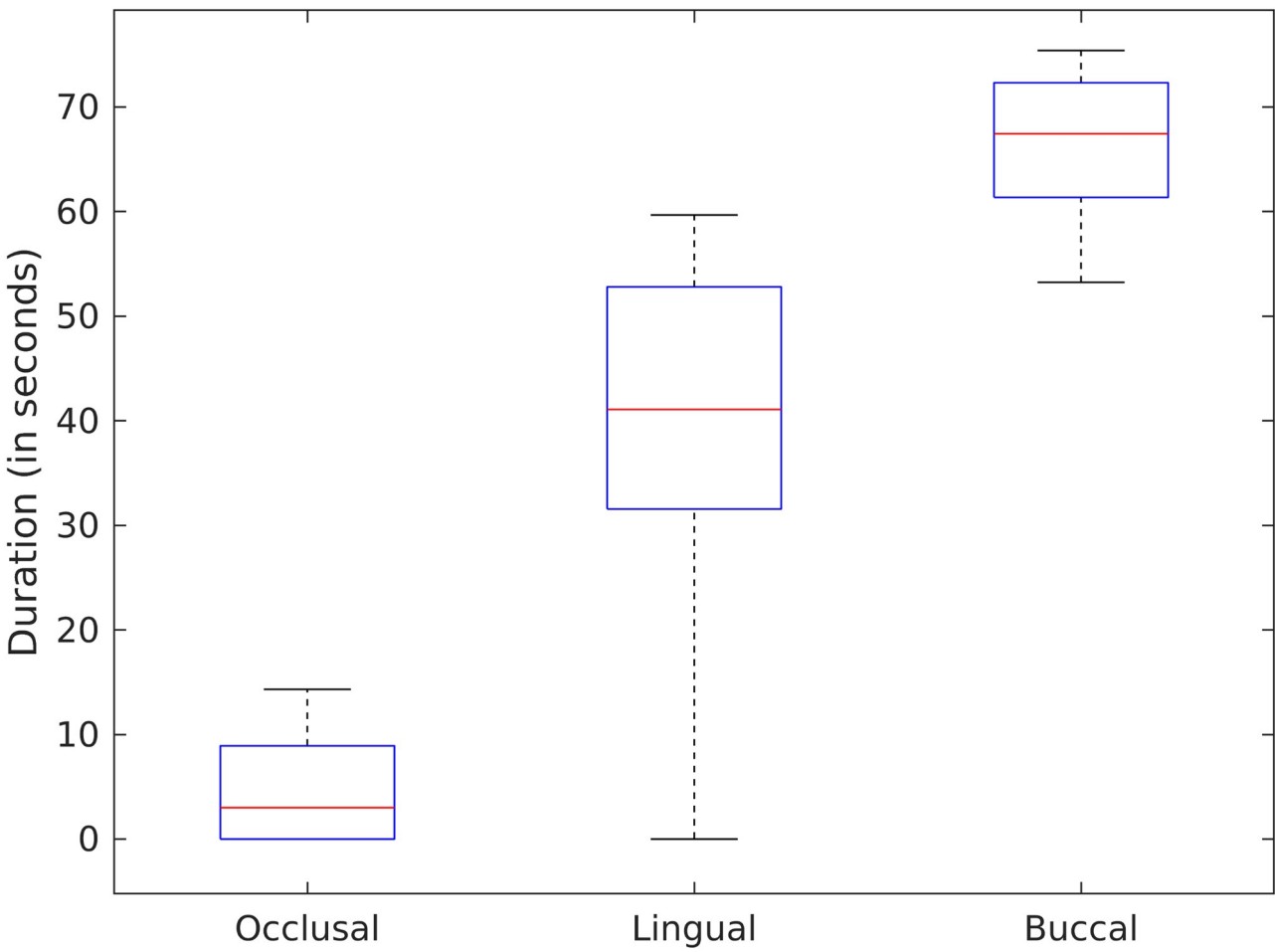

**Fig 6. Within-individual variability in brushing duration for dental surfaces.** Participant 2's brushing duration for the lingual surfaces varied greatly across sessions.

if a dental region is neglected, the plaque removal will be incomplete. The variability in brushing times and dental region coverage may explain some of the considerable heterogeneity in plaque removal reported by the 2014 Cochrane meta-analyses of powered toothbrushes [5].

Our observation that participants tended to brush tooth surfaces unequally substantiates other studies [16,18–20]. Whereas some studies found that lingual surfaces are brushed significantly less than both occlusal and buccal surfaces [16,19], we did not find any statistically significant difference between the times spent brushing the occlusal and lingual surfaces. Also, participants varied in the epochs of excessive brushing pressure and the regions to which they were applied. In general, the occlusal surfaces were more likely to be brushed with excessive pressure. Only a subset (16.7%) of our participants exerted excessive brushing pressures and these were mostly transient (~1 second). This finding contrasts Janusz et al. [21] who reported that 46.3% of their participants exerted excess brushing pressures for more than four seconds within a two-minute brushing session.

Our technology-facilitated study has several strengths. Unlike previous observational studies with limited external validity, our naturalistic study is more representative of real-world brushing behaviors and patterns. We have previously shown that our data collection infrastructure (ROBAS) has a high accuracy for measuring oral health behaviors and can passively

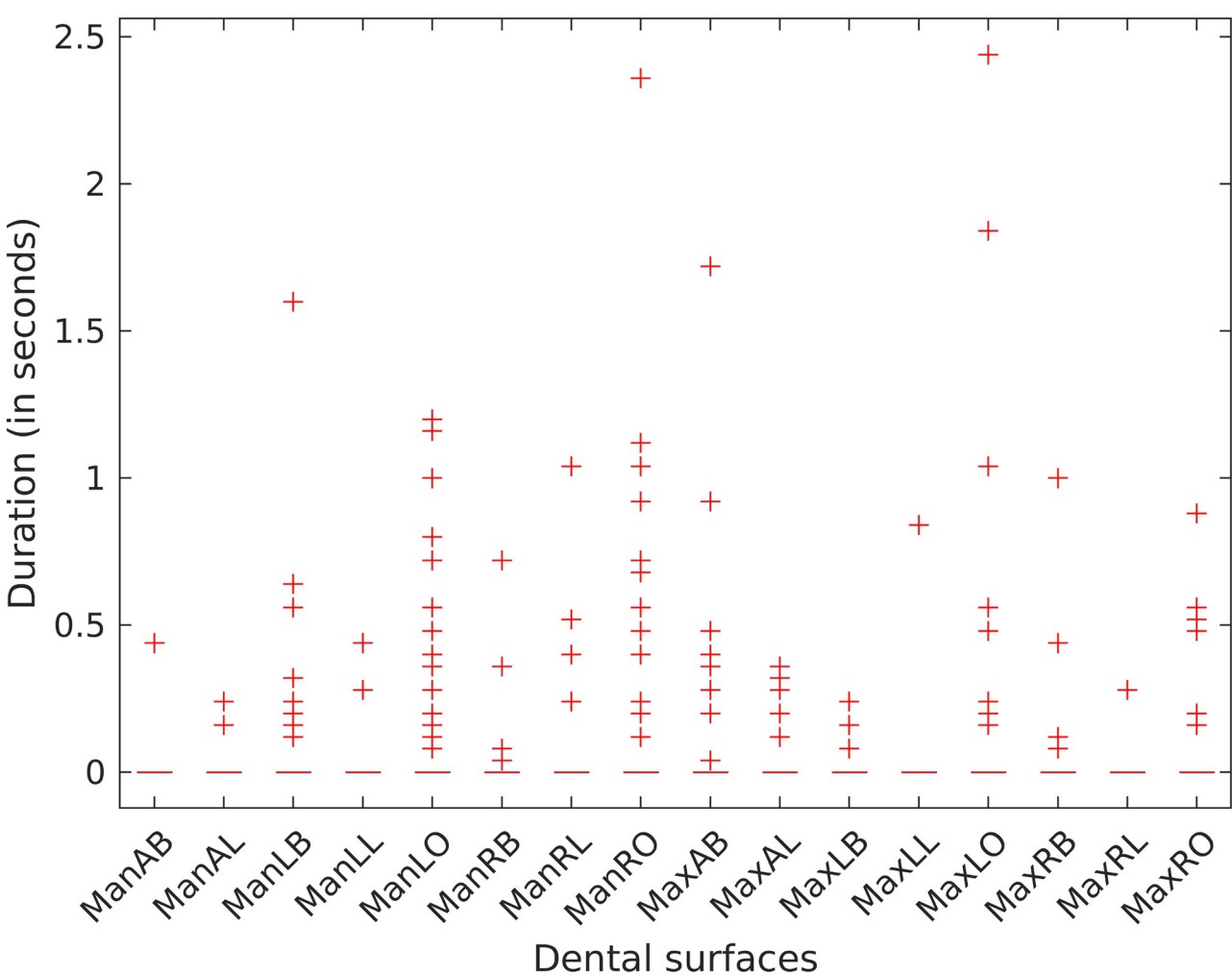

**Fig 7. Episodes of excessive brushing pressure by region.** Occlusal surfaces were most frequently brushed with excessive pressure.

and reliably capture brushing behaviors in the home setting for extended periods [9]. The unobtrusive data collection helps distinguish between person-to-person variability and within-person session-to-session variability via a repeated measurement study design. Anchored by corresponding video recordings of brushing sessions, the data from the brush sensors can be used to accurately reconstruct brushing motions and infer coverage of a dental region. Furthermore, it allows us to ignore pauses and transitions of the brush head and focus on actual brushing times when the brush was in contact with tooth surface. The ecologically-valid measurement of brushing efficiency and the finding of inconsistent brushing patterns set the stage for personalized feedback on brushing patterns. Thus, an individual neglecting to brush a dental quadrant for at least 30 seconds or ignoring certain dental regions could be provided individually tailored feedback and behavioral nudges to improve brushing efficiency. Study limitations include the fact that the study participants were recruited as a convenience sample and involved a young well-educated and dentally-aware group of college students. This may limit the generalizability of our findings to an older and more socioeconomically diverse population.

In summary, our study reveals that users of powered toothbrushes vary substantially in their application of powered toothbrushes and diverge from recommended brushing practices. The inconsistent brushing patterns, between and within individuals, may result in inefficient brushing behaviors.

Our findings underscore the limited uptake of generic oral self-care recommendations and emphasize the need for personalized brushing recommendations that derive from the objective sensor data provided by powered toothbrushes. Simply asking individuals to brush longer or more frequently may not result in a more thorough brushing behaviors.

## Supporting information

**S1 Appendix. Statistical analysis.**
(DOCX)

## Acknowledgments

The authors wish to acknowledge the material support provided Oral-B/Procter & Gamble. The content is solely the responsibility of the authors and does not necessarily represent the official views of the National Institutes of Health or Procter & Gamble.

## Author Contributions

**Conceptualization:** Mahmoud Essalat, Vivek Shetty.

**Data curation:** Mahmoud Essalat, Sumukh Kak, E. Jun Chang, Isabel Roig Penso, Rachel J. Kulchar.

**Formal analysis:** Mahmoud Essalat, Douglas Morrison.

**Funding acquisition:** Vivek Shetty.

**Methodology:** Mahmoud Essalat, Douglas Morrison.

**Project administration:** Mahmoud Essalat, Vivek Shetty.

**Resources:** Vivek Shetty.

**Software:** Mahmoud Essalat, Douglas Morrison.

**Supervision:** Oscar Hernan Madrid Padilla, Vivek Shetty.

**Validation:** Mahmoud Essalat, Douglas Morrison, Oscar Hernan Madrid Padilla.

**Visualization:** Mahmoud Essalat, Douglas Morrison, Isabel Roig Penso, Rachel J. Kulchar.

**Writing – original draft:** Mahmoud Essalat, Vivek Shetty.

**Writing – review & editing:** Mahmoud Essalat, Douglas Morrison, Sumukh Kak, E. Jun Chang, Isabel Roig Penso, Rachel J. Kulchar, Oscar Hernan Madrid Padilla, Vivek Shetty.

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
