## [Decision Letter · Decision Letter 0]

7 Mar 2022

PONE-D-22-02067A naturalistic study of brushing patterns using electric toothbrushesPLOS ONE

Dear Dr. Mahmoud Essalat,

Thank you for submitting your manuscript to PLOS ONE. After careful consideration, we feel that it has merit but does not fully meet PLOS ONE’s publication criteria as it currently stands. Therefore, we invite you to submit a revised version of the manuscript that addresses the points raised during the review process.

We look forward to receiving your revised manuscript.

Kind regards,

Tanay Chaubal

Academic Editor

PLOS ONE

Journal Requirements:

Additional Editor Comments:

Dear Dr Mahmoud Essalat,

I congratulate you on writing a manuscript on a novel concept.

However, there are some comments from the reviewers. Kindly address those comments.

Thank you.

Reviewers' comments:

Reviewer's Responses to Questions

**Comments to the Author**

1. Is the manuscript technically sound, and do the data support the conclusions?

Reviewer #1: No

Reviewer #2: Yes

2. Has the statistical analysis been performed appropriately and rigorously? 

Reviewer #1: I Don't Know

Reviewer #2: Yes

3. Have the authors made all data underlying the findings in their manuscript fully available?

Reviewer #1: Yes

Reviewer #2: Yes

4. Is the manuscript presented in an intelligible fashion and written in standard English?

Reviewer #1: Yes

Reviewer #2: Yes

5. Review Comments to the Author

Reviewer #1: The authors have conducted a study on 12 participants to study their natural brushing habits using a rotation oscillation toothbrush. This study attempts to aggregate the data that is available when participants use these brushes at home rather than a professional setting. Although this is a strength of this study, the study does not appear to provide sufficient aggregated statistical data (although it appears that it was done) to substantiate the results and conclusions from the study. The study also might be under powered with only 12 participants (the authors have not provided how the sample size was arrived at).

GENERAL COMMENTS:

The authors use the word electric toothbrushes throughout the study (and title). Since the type of toothbrush studied here was only of the rotation oscillation type, it cannot be generalized to all types of electric toothbrushes. The authors need to use more specific language across the full text and title to correctly represent what is being studied.

In addition, it appears that the participants were video recorded during all sessions of tooth brushing (needs clarification from authors). Although it might be closer to their natural home practices than studying it in a professional setting, it still does not fully resemble a naturalistic brushing patters. Consider rewording the title and abstract/full text information to “at home” rather than naturalistic.

ABSTRACT:

1. Abstract states “analysis of brushing patterns from 120 sessions” – this statement is misleading as it appears that 120 sessions from each participant was studied. Please be more specific that it was 10 per participant and 120 in total.

2. Please use statistical values wherever appropriate when elaborating results. If not statistically significant – please state so.

INTRODUCTION:

1. Page 3, Line 56-58: The authors state that “the considerable unexplained heterogeneity” – the authors have not raised this issue nor are there references to this statement. Please consider identifying key references that support this statement and include a few lines summary to substantiate this statement since this seems to form the central theme of the relevance of this study.

MATERIALS AND METHODS:

1. How was the sample size arrived at? Was this study adequately powered? Please provide information on power calculation

2. Why only 10 out of 50 sessions were used? How were they randomly selected?

3. It is not clear how duration for each sextant surface that was brushed was observed and calculated – please clarify.

4. Were these participants already using electric toothbrushes before? Were they provided any information of how to use them? Did they have any dental diseases? Please include this information as it is pertinent to the study.

5. Although the statistical methods used are described in sufficient detail, the outcome of the analysis is lacking in the paper and are not presented in the appropriate locations to substantiate the important results and conclusions.

RESULTS:

1. Page 8, 9, Line 167- 172: No statistical values are reported to substantiate statements like “substantial variation”. Also, the authors state “estimated variability” – were they not directly measured?

2. Page 9, Line 189-191: No statistical values are reported to substantiate statements.

3. Page 10, Line 196-199: No statistical values are reported to substantiate statements.

DISCUSSION:

1. Page 11, Line 220-222: The authors state “Our study showed that electric toothbrushes are not used optimally for plaque removal.” Plaque removal was not measured as part of this study and this line would be an extrapolation of the results that is best avoided.

2. There are some redundant sentences in the discussion. Specifically, Page 10, 11 Lines 210 – 212 and Page 11 Line 224-225.

3. Page 12, Line 249, 251: Was the high criterion validity measured by the authors in this study?

4. Page 13, Line 260-263: The authors state “dentally aware group.” Please clarify the nature of this population.

FIGURES AND GRAPHS:

Figures requires titles and footnotes.

Reviewer #2: Well constructed study.

1. Please be consistent between the terms "electric" and "powered".

2.Age range and gender have not been mentioned in materials and methods.

3. Why were the participants not given the manufacturer's brushing instructions?

6. PLOS authors have the option to publish the peer review history of their article (what does this mean?). If published, this will include your full peer review and any attached files.

Reviewer #1: No

Reviewer #2: No

---

## [Author Response · Author response to Decision Letter 0]

18 Mar 2022

A naturalistic study of brushing patterns using electric toothbrushes

Essalat, Morrison, Kak, Chang, Roig, Kulchar, Padilla, Shetty

[PONE-D-22-02067] - [EMID:48f78bddccdd4d40]

Responses to the reviewers’ comments

Dear Editor,

Please find enclosed our revised manuscript. We are heartened that our submission was resonant with the reviewers and thank them for their constructive comments. Where appropriate, we have revised the manuscript accordingly and provide our specific responses below.

Reviewer #1 comments:

the study does not appear to provide sufficient aggregated statistical data (although it appears that it was done) to substantiate the results and conclusions from the study. The study also might be under powered with only 12 participants (the authors have not provided how the sample size was arrived at).

We have added a paragraph in materials and methods to clarify this point. Specifically, our observational study was not designed to detect a predetermined difference in measurement between two groups (power). The focus of our study was on clarifying individual differences in terms of toothbrushing duration, dental regions covered, and episodes of excessive brushing pressure in each dental region. 

The authors use the word electric toothbrushes throughout the study (and title). Since the type of toothbrush studied here was only of the rotation oscillation type, it cannot be generalized to all types of electric toothbrushes. The authors need to use more specific language across the full text and title to correctly represent what is being studied.

As suggested, we have changed replace the term” electric” with the more appropriate “powered” toothbrush through the revised manuscript. 

In addition, it appears that the participants were video recorded during all sessions of tooth brushing (needs clarification from authors). Although it might be closer to their natural home practices than studying it in a professional setting, it still does not fully resemble a naturalistic brushing pattern. Consider rewording the title and abstract/full text information to “at home” rather than naturalistic.

While this may be a matter of semantics, naturalistic is the term more commonly used in mHealth research to describe data collected in the natural environments. The “at-home” may be a misnomer in the case of our student participants who conducted the brushing sessions in their student residences/dorms. 

ABSTRACT:

1. Abstract states “analysis of brushing patterns from 120 sessions” – this statement is misleading as it appears that 120 sessions from each participant was studied. Please be more specific that it was 10 per participant and 120 in total.

As suggested, we have corrected that statement to “total 120 sessions” and clarified through greater specificity of the language (10 sessions per participant and 120 in total). 

2. Please use statistical values wherever appropriate when elaborating results. If not statistically significant – please state so.

As suggested, we have added the p-values and confidence intervals. 

INTRODUCTION:

1. Page 3, Line 56-58: The authors state that “the considerable unexplained heterogeneity” – the authors have not raised this issue nor are there any references to this statement. Please consider identifying key references that support this statement and include a few lines summary to substantiate this statement since this seems to form the central theme of the relevance of this study.

As suggested, we have improved the explanation and added the corresponding reference. Specifically, we now state “The considerable heterogeneity in plaque removal with powered toothbrushes (I(2) > 80%) observed in [5], could not be explained by the difference in types of powered toothbrushes, and may more likely be explained by varying toothbrushing patterns.

MATERIALS AND METHODS:

1. How was the sample size arrived at? Was this study adequately powered? Please provide information on power calculation

We used convenience sampling for our study that utilized repeated measures. The sample size was based on previous studies [1, 2] which had recruited 12, 10, and 12 subjects. Given the observational nature of the study of individual differences, no power calculation was performed. We submit that our use of confidence intervals and p-values are sufficient to identify trends and results that were statistically significant. 

2. Why only 10 out of 50 sessions were used? How were they randomly selected?

Annotating videos for machine learning purposes is a very time and resource intensive task. For elucidating individual variability, we choose to focus on annotating 10 sessions out of the of 50 sessions provided by each participant. The sessions for annotation were selected randomly with each brushing session having an equal chance of being selected. The “Data Processing and Annotation” subsection has been updated to reflect this point.

3. It is not clear how duration for each sextant surface that was brushed was observed and calculated – please clarify.

The revised “Data Processing and Annotation” subsection clarifies this now. Specifically, “Every epoch of brushing a dental region that lasted more than 0.5 second was annotated by marking the beginning and end timestamp of that brushing epoch.”

4. Were these participants already using electric toothbrushes before? Were they provided any information of how to use them? Did they have any dental diseases? Please include this information as it is pertinent to the study.

As the text clarifies, the participants were young healthy college students with no dental disease. None had prior experience with powered toothbrushes. Basic instruction on the use of the brush and setting up the data collection system was given and then the participants were instructed to freely brush their teeth in a manner most natural to them (i.e., no structure brushing patterns, such as the Bass method, were prescribed).

5. Although the statistical methods used are described in sufficient detail, the outcome of the analysis is lacking in the paper and are not presented in the appropriate locations to substantiate the important results and conclusions.

As suggested, we have amplified on this in greater detail now (also see response to comment 1 above)

RESULTS:

1. Page 8, 9, Line 167- 172: No statistical values are reported to substantiate statements like “substantial variation”. Also, the authors state “estimated variability” – were they not directly measured?

We added the coefficient of variation and consider it to be substantial since it is larger than 5%. Also, we added this to the section A3 in the supplementary file. We changed the term “estimated” to “observed” since it is empirically measured. 

2. Page 9, Line 189-191: No statistical values are reported to substantiate statements.

We have now included the coefficient of variations for all the regions brushed by all the participants in the supplementary file subsection A2.2. We now state that: “Coefficient of variation for all the regions that were brushed during a brushing session was greater than 20%. (See S1 Appendix A2). “

Also, we now included the p-value and confidence intervals for the claimed observations. 

DISCUSSION:

1. Page 11, Line 220-222: The authors state “Our study showed that electric toothbrushes are not used optimally for plaque removal.” Plaque removal was not measured as part of this study and this line would be an extrapolation of the results that is best avoided.

We agree and have removed that sentence. 

2. There are some redundant sentences in the discussion. Specifically, Page 10, 11 Lines 210 – 212 and Page 11 Line 224-225.

We removed the redundancies.

3. Page 12, Line 249, 251: Was the high criterion validity measured by the authors in this study?

We have removed that sentence.

4. Page 13, Line 260-263: The authors state “dentally aware group.” Please clarify the nature of this population.

We have now clarified them as college students. 

FIGURES AND GRAPHS:

Figures requires titles and footnotes.

We were following PLOS ONE guidelines that instruct authors, “Do not include author names, article title, or figure number/title/caption within figure files. That information will go into your figure caption in the manuscript.” 

Reviewer #2 comments:

1. Please be consistent between the terms "electric" and "powered".

As suggested, electric has been replaced with powered.

2. Age range and gender have not been mentioned in materials and methods.

This is now included in the Materials and Methods section. 

3. Why were the participants not given the manufacturer's brushing instructions?

There are no specific manufacturers’ instructions. However, participants were instructed on how to set up the brush and data collection equipment- they then brushed using their natural preferences. Details are now provided in the Materials and Methods section.

References:

[1] Huang H, Lin S. Toothbrushing monitoring using wrist watch. In Proceedings of the 14th ACM Conference on Embedded Network Sensor Systems CD-ROM 2016 Nov 14 (pp. 202-215).

[2] Luo C, Feng X, Chen J, Li J, Xu W, Li W, Zhang L, Tari Z, Zomaya AY. Brush like a dentist: accurate monitoring of toothbrushing via wrist-worn gesture sensing. In IEEE INFOCOM 2019-IEEE Conference on Computer Communications 2019 Apr 29 (pp. 1234-1242). IEEE.

[3] Lee YJ, Lee PJ, Kim KS, Park W, Kim KD, Hwang D, Lee JW. Toothbrushing region detection using three-axis accelerometer and magnetic sensor. IEEE Transactions on Biomedical Engineering. 2011 Dec 22;59(3):872-81.

---

## [Decision Letter · Decision Letter 1]

5 Apr 2022

PONE-D-22-02067R1A naturalistic study of brushing patterns using powered toothbrushesPLOS ONE

Dear Dr. Mahmoud Essalat,

Thank you for submitting your manuscript to PLOS ONE. After careful consideration, we feel that it has merit but does not fully meet PLOS ONE’s publication criteria as it currently stands. Therefore, we invite you to submit a revised version of the manuscript that addresses the points raised during the review process.

Kindly address the comments mentioned by the reviewers.

We look forward to receiving your revised manuscript.

Kind regards,

Tanay Chaubal

Academic Editor

PLOS ONE

Journal Requirements:

Reviewers' comments:

Reviewer's Responses to Questions

**Comments to the Author**

1. If the authors have adequately addressed your comments raised in a previous round of review and you feel that this manuscript is now acceptable for publication, you may indicate that here to bypass the “Comments to the Author” section, enter your conflict of interest statement in the “Confidential to Editor” section, and submit your "Accept" recommendation.

Reviewer #1: (No Response)

Reviewer #2: All comments have been addressed

2. Is the manuscript technically sound, and do the data support the conclusions?

Reviewer #1: Partly

Reviewer #2: Yes

3. Has the statistical analysis been performed appropriately and rigorously? 

Reviewer #1: Yes

Reviewer #2: Yes

4. Have the authors made all data underlying the findings in their manuscript fully available?

Reviewer #1: Yes

Reviewer #2: Yes

5. Is the manuscript presented in an intelligible fashion and written in standard English?

Reviewer #1: Yes

Reviewer #2: Yes

6. Review Comments to the Author

Reviewer #1: Some comment(s) have not been addressed (the comment says it has been).

Specifically:

1. There are some redundant sentences in the discussion. Specifically, Page 10, 11

Lines 210 – 212 and Page 11 Line 224-225.

Reviewer #2: Well written manuscript. suitable for acceptance. the concept of the manuscript is novel and gives an insight into the brushing pattern of subjects who have had no directions on how to brush.

7. PLOS authors have the option to publish the peer review history of their article (what does this mean?). If published, this will include your full peer review and any attached files.

Reviewer #1: No

Reviewer #2: No

---

## [Author Response · Author response to Decision Letter 1]

12 Apr 2022

Dear Editor,

I am delighted to learn the favorable reaction of the reviewers.

Allow me to clarify the residual issue of redundant sentences (Critique 6- Reviewer 1).

There are some redundant sentences in the discussion. Specifically, Page 10, 11

Lines 210 – 212 and Page 11 Line 224-225.

In writing the discussion, we followed the general guidelines provided by PLOS (https://plos.org/resource/how-to-write-conclusions/).

The first paragraph of the discussion section provides a high-level summary of our main findings. The subsequent paragraphs begin with a brief description of each finding, followed by a brief interpretation of the finding and a comparison/contrast with findings by other investigators.

---

## [Decision Letter · Decision Letter 2]

4 May 2022

A naturalistic study of brushing patterns using powered toothbrushes

PONE-D-22-02067R2

Dear Dr. Mahmoud Essalat,

We’re pleased to inform you that your manuscript has been judged scientifically suitable for publication and will be formally accepted for publication once it meets all outstanding technical requirements.

Kind regards,

Tanay Chaubal

Academic Editor

PLOS ONE

Additional Editor Comments (optional):

Reviewers' comments:

Reviewer's Responses to Questions

**Comments to the Author**

1. If the authors have adequately addressed your comments raised in a previous round of review and you feel that this manuscript is now acceptable for publication, you may indicate that here to bypass the “Comments to the Author” section, enter your conflict of interest statement in the “Confidential to Editor” section, and submit your "Accept" recommendation.

Reviewer #1: (No Response)

2. Is the manuscript technically sound, and do the data support the conclusions?

Reviewer #1: Partly

3. Has the statistical analysis been performed appropriately and rigorously? 

Reviewer #1: Yes

4. Have the authors made all data underlying the findings in their manuscript fully available?

Reviewer #1: Yes

5. Is the manuscript presented in an intelligible fashion and written in standard English?

Reviewer #1: Yes

6. Review Comments to the Author

Reviewer #1: (No Response)

7. PLOS authors have the option to publish the peer review history of their article (what does this mean?). If published, this will include your full peer review and any attached files.

Reviewer #1: No

---

## [Editor Report · Acceptance letter]

11 May 2022

PONE-D-22-02067R2 

A naturalistic study of brushing patterns using powered toothbrushes 

Dear Dr. Essalat:

I'm pleased to inform you that your manuscript has been deemed suitable for publication in PLOS ONE. Congratulations! Your manuscript is now with our production department. 

Kind regards, 

on behalf of

Dr. Tanay Chaubal 

Academic Editor

PLOS ONE